# First-step Inference in Diffusion Models Learns Image De-whitening

## Abstract

Diffusion models have emerged as powerful generative models for image synthesis, yet the intricate relationship between input noise and generated images remains not fully understood. In this paper, we investigate the correlation between noise and images generated through deterministic DDIM sampling, uncovering fundamental elements that are present across different diffusion models. More specifically, we demonstrate that a one-step approximation of the mapping learned by these models closely relates to Zero-phase Component Analysis (ZCA) inverse whitening transform, which maximizes the correlation between source and target distributions. We leverage this insight to develop a simple and yet effective model-agnostic method for sampling correlated noises and showcase applications for image variation generation and editing.

## 1 Introduction

The landscape of generative artificial intelligence has witnessed an unprecedented evolution, notably marked by the stellar advance of diffusion models (Sohl-Dickstein et al., 2015; Ho et al., 2020). Diffusion models represent the state of the art in terms of their generative capabilities, being used for a myriad of generative tasks, ranging from text-to-image (Rombach et al., 2022) and text-to-video (Ho et al., 2022; Villegas et al., 2022) models to sound guided (Alexanderson et al., 2022) and text-based (Tevet et al., 2022) motion synthesis. Diffusion models operate on a straightforward but powerful principle: reversing a diffusion process that progressively corrupts images into noise. Due to its maximum entropy characteristic, Gaussian noise is commonly employed to train diffusion models, since it allows the network to approximate image distributions in an unbiased way.

In this paper, we investigate the relationship between noise samples and images produced by diffusion models. We are motivated by the observation that diffusion models sampled by deterministic DDIM (Song et al., 2020) inherently preserve and translate correlations present in noise samples to images (Khrulkov et al., 2022). Figure 2 illustrates this observation: by progressively averaging distinct noise samples and comparing with averages of their corresponding images, we demonstrate that visual structures on these averages are spatially similar.

To further understand this relationship, we propose a fixed point diffusion strategy that approximates the mapping between a noise sample and a generated image in a single-step. Surprisingly, this single-step mapping is a remarkable approximation of a "de-whitening" operation, which is computed by the inverse of the ZCA whitening transform (Bell & Sejnowski, 1997). Furthermore, we show that a ZCA whitening matrix that is independently computed for the ImageNet dataset (Deng et al., 2009) is remarkably close to the least-squares fitting between noise-image pairs of our single-step diffusion approximation.

With the understanding that de-whitening is a fundamental element of diffusion models, we are able to create an efficient optimization algorithm that generates initial noise samples that produce similar images *across completely different diffusion models* (Figures 1 and 9). To the best of our knowledge, our work is the first one to solve this inverse problem, resulting in model-agnostic noise samples that generate images with a specific content. Furthermore, this relationship can be explored to improve results of popular image editing approaches such as SDEdit (Meng et al., 2022) (Figure 10). In summary, our work makes the following key contributions:

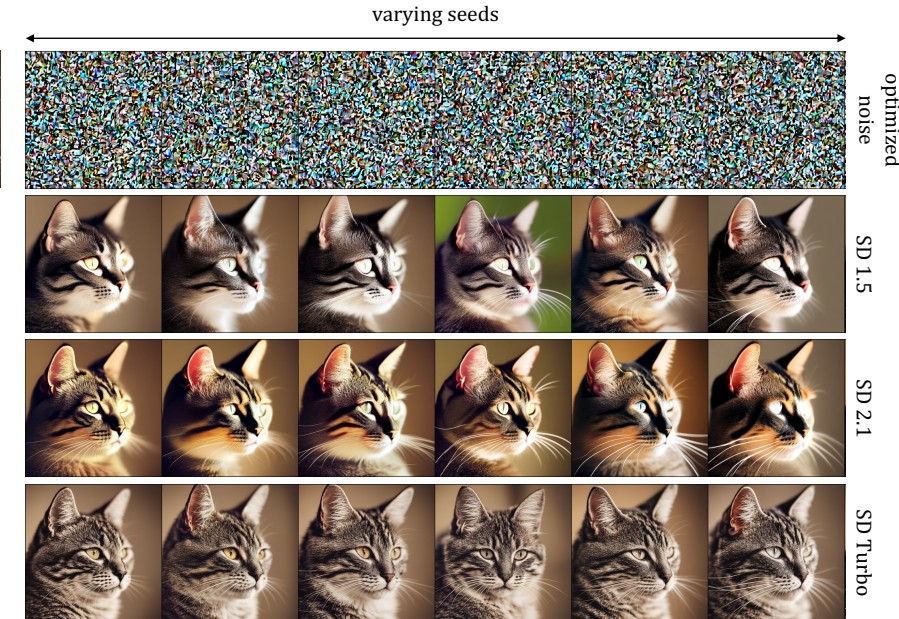

Figure 1: **Model-agnostic image variation generation.** We uncover the existence of high correlation between image and noise in diffusion-based methods. Variations of an image that preserves the global structure can be generated by simply searching for noises that are correlated with the target image. The search for suitable noises *does not involve any diffusion models* and the same resulting noise can be used across different models sharing the same input space.

- We uncover the existence of a strong correlation between noise samples and their corresponding images in diffusion models.
- We demonstrate that this correlation can be efficiently analyzed through a single-step approximation of the noise-image mapping learned by diffusion models. This single-step approximation results in a mapping that is close to the ZCA de-whitening transform on the noise distribution (Section 3).
- Leveraging these insights, we propose a novel simulated annealing optimization for model-agnostic noise inversion, creating a new avenue for image generation and manipulation (Section 5).

## 2 PRELIMINARIES

### 2.1 DIFFUSION MODELS

Diffusion models (Sohl-Dickstein et al., 2015; Ho et al., 2020) generate image samples through a sequential process of denoising an initially Gaussian-sampled noise. Gaussian noise $\boldsymbol{\epsilon}_t \sim \mathcal{N}(\mathbf{0}, \mathbf{I})$ is subsequently added to a clean image $\mathbf{z}_0 \sim p_{\text{data}}$ during the training of diffusion models:

$$\mathbf{z}_t = \sqrt{\bar{\alpha}_t}\,\mathbf{z}_0 + \sqrt{1 - \bar{\alpha}_t}\,\boldsymbol{\epsilon}_t, \tag{1}$$

where $\alpha_t$ defines a fixed noise schedule and $\bar{\alpha}_t = \prod_{s=0}^{t} \alpha_s$ at timestep $t$.

Once the diffusion model is trained, a sequence of denoising steps is used to sample images from the model. The denoising step often relies on classifer-free guidance (CFG) (Ho et al., 2020) to generate high-quality samples. CFG linearly interpolates a text-conditioned denoising step and an unconditional one:

$$\hat{\boldsymbol{\epsilon}}_t = \boldsymbol{\epsilon}_\theta(\mathbf{z}_t; t) + \omega(\boldsymbol{\epsilon}_\theta(\mathbf{z}_t; t, c) - \boldsymbol{\epsilon}_\theta(\mathbf{z}_t; t)), \tag{2}$$

where $\boldsymbol{\epsilon}_\theta$ is the trained diffusion model, and $\omega$ is the classifier-free guidance scale. The revised denoising prediction $\hat{\boldsymbol{\epsilon}}_t$ is then used to update the noisy image $\mathbf{z}_t$, and estimate the clean $\mathbf{z}_{0|t}$ image

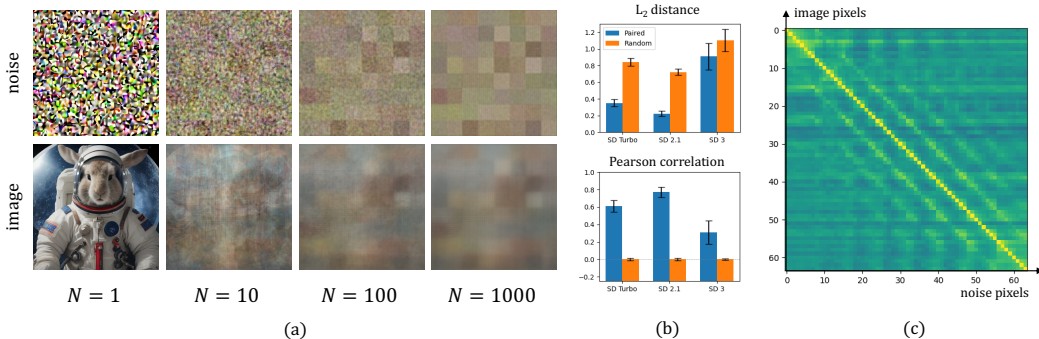

Figure 2: **Correlation between noise samples and generated images.** (a) The left-most column shows a comparison between the noise seed (top) and the result (bottom) of SDXL-Turbo evaluated with DDIM sampling with the corresponding seed ($64 \times 64$). This seed is downsampled ($8 \times 8$) to extract its low frequency component. We then sample 1000 different noise maps with this same low frequency component, which are then combined with random prompts to generate 1000 images. From left to right, we progressively average more images and noise seeds. Top row shows averaged noises (and the shared low frequency component), while bottom row shows averaged images. (b) Computing the average L2 distance and Pearson correlation between paired noises and images shows a much stronger correlation then with randomly sampled noises. (c) The cross-correlation between paired image and noise pixels shows strong response on the diagonal, suggesting a local, per-pixel correlation.

for timestep $t$ with the Tweedies formulation (Robbins, 1992) as

$$\mathbf{z}_{0|t} = (\mathbf{z}_t - \sqrt{1 - \bar{\alpha}_t}\hat{\boldsymbol{\epsilon}}_t)/\sqrt{\bar{\alpha}_t}. \tag{3}$$

Various sampling schemes are designed for efficiency (Ho et al., 2020; Song et al., 2020; Salimans & Ho, 2022). In this paper, we adopt deterministic DDIM (Song et al., 2020) defined by

$$\mathbf{z}_{t-1} = \sqrt{\bar{\alpha}_{t-1}}\,\mathbf{z}_{0|t} + \sqrt{1 - \bar{\alpha}_{t-1} - \sigma_t^2}\,\hat{\boldsymbol{\epsilon}}_t + \sigma_t\boldsymbol{\epsilon}_t, \tag{4}$$

where $\sigma_t$ controls the stochasticity of the sampling process and $\boldsymbol{\epsilon}_t \sim \mathcal{N}(\mathbf{0}, \mathbf{I})$.

## 2.2 Image whitening

Image *whitening* or *sphering* (Bell & Sejnowski, 1997; Kessy et al., 2018) refers to the linear operation that transforms a random vector $\mathbf{x} \in \mathbb{R}^D$ of mean $\boldsymbol{\mu}$ and covariance matrix $\mathbb{E}[\mathbf{x}\mathbf{x}^T] = \boldsymbol{\Sigma}$ to a new random vector

$$\mathbf{z} = \mathbf{W}(\mathbf{x} - \boldsymbol{\mu})$$

such that $\mathbf{z}$ has zero mean and unit diagonal covariance $\mathbb{E}[\mathbf{z}\mathbf{z}^T] = \mathbf{I}$. The $D \times D$ matrix $\mathbf{W}$ is called the *whitening matrix*. The objective is to remove correlations between features and to normalize variances. It is a common preprocessing technique particularly useful in computer vision tasks, dimensionality reduction, and feature extraction. In the context of images, the dimension of the vector is the number of pixels $D = h \times w \times c$. While it can be shown that the whitening matrix should satisfy

$$\mathbf{W}^\top \mathbf{W} = \boldsymbol{\Sigma}^{-1},$$

this does not determine the whitening matrix uniquely, but only up to a mulitplication by an orthogonal matrix. Different whitening matrices have different properties, we refer to Kessy et al. (2018) for a detailed overview of the topic and review only the three most common ones:

$$\mathbf{W}_{\text{PCA}} = \boldsymbol{\Lambda}^{-1/2}\mathbf{U}^T, \qquad \mathbf{W}_{\text{ZCA}} = \mathbf{U}\boldsymbol{\Lambda}^{-1/2}\mathbf{U}^T, \qquad \mathbf{W}_{\text{Cholesky}} = \mathbf{L}^\top. \tag{5}$$

where $\boldsymbol{\Sigma} = \mathbf{U}\boldsymbol{\Lambda}\mathbf{U}^\top = \mathbf{L}\mathbf{L}^\top$ are respectively the SVD and Cholesky decomposition of the data covariance matrix.

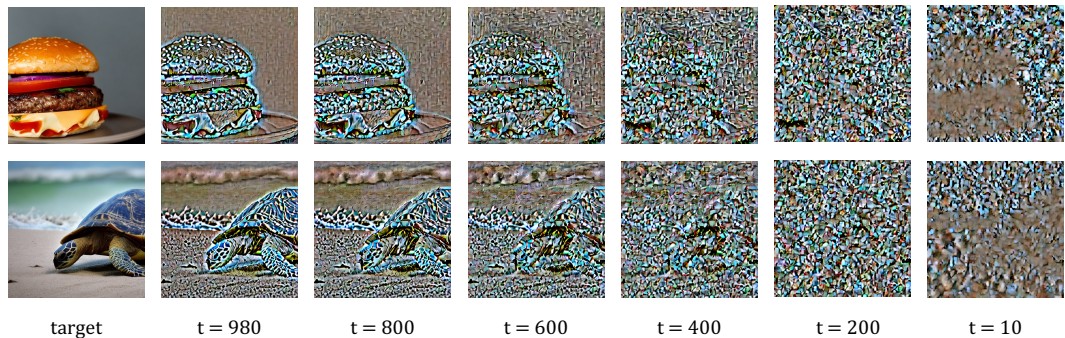

| target | $t = 980$ | $t = 800$ | $t = 600$ | $t = 400$ | $t = 200$ | $t = 10$ |

Figure 3: Performing our fixed-point iteration to recover the perfect denoising noise for different times of the denoising process reveals how the model removes the noise in a non-uniform manner.

**PCA Whitening.** PCA (Principal Component Analysis) whitening (Friedman, 1987) utilizes the eigenvectors of the covariance matrix to decorrelate the data. PCA whitening effectively rotates the data to align with the principal components.

**ZCA Whitening.** While conceptually similar to PCA whitening, Zero-phase Component Analysis (ZCA) whitening includes an additional rotational step that ensures the whitened data remains as close as possible to the original data in the input space, maximizing the cross-covariance between both distributios. It preserves the geometric structure of the data, making it particularly advantageous for tasks in image processing where retaining the original orientation of the data is important.

**Cholesky Whitening.** Another widely known method is Cholesky whitening, which employs the Cholesky decomposition of the inverse covariance matrix.

## 3  SINGLE-STEP MAPPING VIA FIXED POINT ITERATION

We are interested in understanding the behavior of diffusion models when the mapping between a noise sample $\epsilon$ and a given image $\mathbf{z}_0$ is defined through a linear operation $\mathbf{z}_0 = \mathbf{A}\epsilon$. For given state $\mathbf{z}_t$ (inset image) and its corresponding model denoising prediction $\epsilon_\theta(\mathbf{z}_t, t)$, a mapping that connects this state to a sample closer to the image distribution $\mathbf{z}_{0|t}$, can be computed through Equation (3). When $t$ is large, the predictions $\mathbf{z}_{0|t}$ will usually produce images that are blurry and are far way from the manifold of the data distribution. Instead of 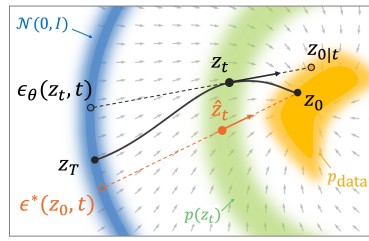 fixing $\mathbf{z}_t$, we seek a noise sample $\epsilon$ that directly connects the given image $\mathbf{z}_0$ to a new state $\hat{\mathbf{z}}_t$ (on the same manifold as $\mathbf{z}_t$) through a spherical interpolation (red dashed line). This can be achieved by a minimization problem:

$$\min_{\epsilon} \|\epsilon_\theta(\sqrt{\bar{\alpha}_t}\,\mathbf{z}_0 + \sqrt{1 - \bar{\alpha}_t}\,\epsilon;\, t) - \epsilon\|_2^2. \tag{6}$$

The minimized energy in Equation (6) can be seen as the standard denoising score matching objective (Ho et al. (2020); Poole et al. (2022); Kim et al. (2024)) with two modifications: the timestep $t$ is fixed, and the typical Gaussian noise added to the clean image $\mathbf{z}_0$ is replaced by an optimizable variable $\epsilon$. This tight connection with the diffusion training loss suggests that the solution to our minimization problem is close to a random Gaussian noise sample. However, we note that whether it is close to Gaussian noise sample is dependent on the timestep; this is visualized by obtaining noise maps for varying timestep values (Figure 3). Even more surprisingly, we found that solutions to the optimization problem when $t$ is large share striking similarities with the result of applying a ZCA whitening operation to the input images (Figure 4). We investigate this finding further in Section 4. .

Equation (6) is solved when the noise predicted by the network at $t$ is equal to the noise added to $\mathbf{z}_0$, thus effectively denoising into the target image in a single step starting from $t$. One can

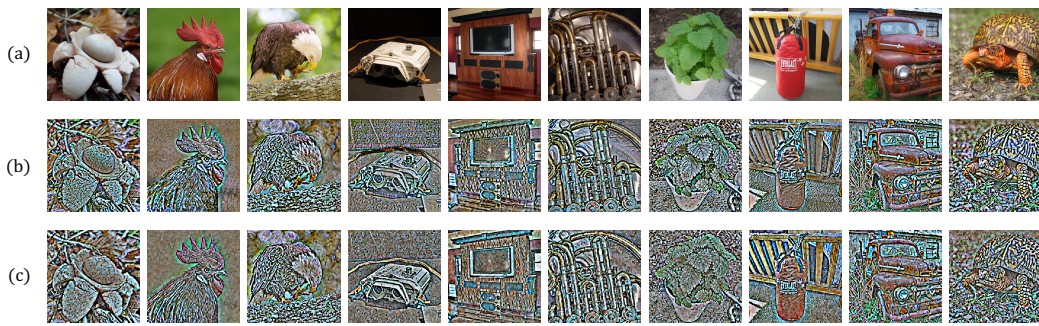

Figure 4: **Solutions to the minimization problem in Equation (6) share striking resemblance to applying ZCA whitening to the original images when** $t \simeq T$**.** By solving for a noise map (b) that would generate an image (a) in a single inference step, we obtain images that look similar to a ZCA-whitened version of the original image (c).

efficiently solve it through a fixed-point iteration (Pan et al., 2023; Samuel et al., 2024), where the noise candidate at iteration $n + 1$ is computed as

$$\boldsymbol{\epsilon}^{n+1} = \boldsymbol{\epsilon}_\theta(\sqrt{\bar{\alpha}_t}\, \mathbf{z}_0 + \sqrt{1 - \bar{\alpha}_t}\, \boldsymbol{\epsilon}^n; t). \tag{7}$$

Equation (7) works by recursively evaluating the network prediction at a *fixed time* $t$, which yields the solution $\boldsymbol{\epsilon}^*(\mathbf{z}_0, t)$. Since denoising networks in diffusion models are trained to predict the amount of noise at a certain level, continuously updating $\boldsymbol{\epsilon}^{n+1}$ until convergence effectively solves Equation (6).

## 4    FIRST-STEP INFERENCE & ZCA WHITENING

Figure 4 shows the comparison between the optimized images from Equation (6) and images obtained through ZCA whitening. The notable resemblance can be surprising for several reasons. Firstly, it suggests that the single-step mapping between noise and image using a diffusion model is actually a linear transformation. While this does not mean the actual full mapping is linear, it still provides valuable insight on the global structure of the learned mapping between the two distributions. Secondly, ZCA whitening is only dependent on the data distribution. Thus, the connection with ZCA would imply that the first step of the diffusion model does not depend on the model, but only on the training distribution. To go beyond simple visual assessment, we propose to experimentally validate the following hypothesis.

**Hypothesis 1**    *At high noise levels, the solution to Equation (6) can be linearly approximated by a ZCA whitening operation, i.e.,*

$$\boldsymbol{\epsilon}^*(\mathbf{z}_0, t) \simeq \mathbf{W}_{\text{ZCA}}\mathbf{z}_0, \qquad \text{for} \quad t \simeq T, \quad \mathbf{z}_0 \sim p_{\text{data}}. \tag{8}$$

### 4.1    EXPERIMENTAL SETUP

We consider $N = 50000$ images from the validation set of ImageNet (Deng et al., 2009), and generate paired noise and image data by optimizing Equation (7) for each sample using $t = 0.98T$, similar to Figure 4 (a) and (b). We reshape the dataset into two matrices $\mathbf{X}$ (images) and $\mathbf{Z}$ (noise) of dimensions $N \times D$, where $D$ denotes the number of pixels and channels, and subsequently center them. We assume these pairs relate linearly through an unknown matrix $\mathbf{W}_{\text{Diff}}$ of size $D \times D$. Therefore, we can estimate it using a least squares formulation, which minimizes the energy

$$\mathbf{W}_{\text{Diff}} = \arg\min_{\mathbf{W}} ||\mathbf{X}\mathbf{W}^\top - \mathbf{Z}||_2^2.$$

We solve this using the normal equations as:

$$\mathbf{W}_{\text{Diff}}^\top = (\mathbf{X}^\top \mathbf{X})^{-1}\mathbf{X}^\top \mathbf{Z} \quad \Longleftrightarrow \quad \mathbf{W}_{\text{Diff}} = \mathbf{Z}^\top \mathbf{X}(\mathbf{X}^\top \mathbf{X})^{-\top} \tag{9}$$

Concurrently, we also use the images $\mathbf{X}$ to compute several standard whitening matrices including ZCA, ZCA-cor, PCA, PCA-cor, and Cholesky (Kessy et al., 2018) for comparison. The validation of the hypothesis relies on proving three points:

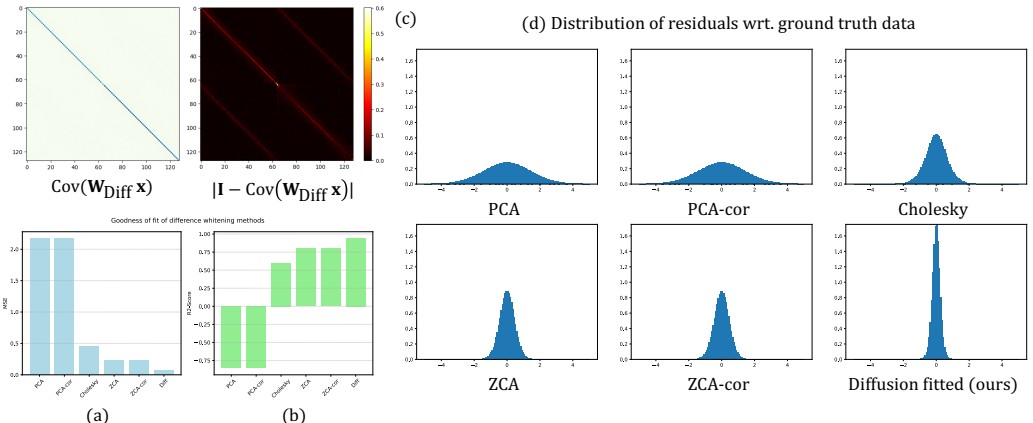

Figure 5: **Experimental analysis of the mapping between images and corresponding solutions of Equation (6).** The MSE (a) and $R^2$ scores (b) show that the ZCA whitening matrix is the second best fit to our noise/image pairs, after the solution of least squares fitting.

- **Linearity**: the linear assumption fits the observed data well
- **Whitening**: the found transformation actually whitens the input images
- **Relationship with ZCA**: the transformation is close to the ZCA whitening matrix

## 4.2 RESULT & ANALYSIS

**Linearity.** We measure the goodness of fit of our linear model using the Mean Squared Error (MSE) and the R2-score:

$$\mathrm{MSE}(\mathbf{X}, \mathbf{Z}) = \frac{1}{ND}||\mathbf{X}\mathbf{W}^\top - \mathbf{Z}||_2^2 \qquad R^2(\mathbf{X}, \mathbf{Z}) = 1 - \frac{\mathrm{MSE}(\mathbf{X}, \mathbf{Z})}{\mathrm{Var}(\mathbf{Z})}$$

While small values of MSE indicates a better fit on the data, the scale of the value is hard to assess. The $R^2$-score measures the error of the fit relative to the variance of the data, giving a more interpretable measure. We plot these metrics for the different whitening matrices in Figure 5(a) and 5(b). Unsurprisingly, the one fitted on the data (Diff) achieves the lowest MSE and highest $R^2$ score. Additionally, the distribution of the residual is centered and relatively close to 0, as shown in Figure 5(d). This indicates that the linear assumption is valid, i.e. the first-step inference mapping is close to a linear transformation.

**Whitening.** We verify that the fitted model indeed whitens the images by looking at the covariance matrix of the transformed data $\mathbf{Z}_{\mathrm{Diff}} = \mathbf{X}\mathbf{W}_{\mathrm{Diff}}^\top$. For whitened data, the covariance matrix is close to the identity matrix $\mathbf{I}$. In Figure 5(c), we plot a subset of the covariance matrix for visualization and show the difference with the identity. The low error values confirms that the resulting matrix is close to a whitening matrix.

**Relationship with ZCA.** Lastly, we confirm that the closest whitening transform that explains our paired data is the ZCA whitening. Figure 5(b) shows that the ZCA whitening also has a high $R^2$ score on the dataset, while achieving the lowest norms among known whitening methods (see inset).

| Method | 2-norm | Frob. |
|---|---|---|
| PCA | 7.8795 | 434.2 |
| ZCA | **3.8305** | **136.6** |
| Cholesky | 4.8027 | 204.6 |

Table 1: Comparison of methods using 2-norm and Frobenius norms.

We hypothetize that the gap between the fitted one (Diff) and (ZCA) might be partly due to the fact that the ZCA whitening matrix was only estimated on a subset of ImageNet, while the fitted one would reflect the entire training distribution of the diffusion model.

**Analysis.** We have proved experimentally that the first-step inference of diffusion models approximates a ZCA *de*-whitening transform (as the mapping from image to noise is a whitening operation).

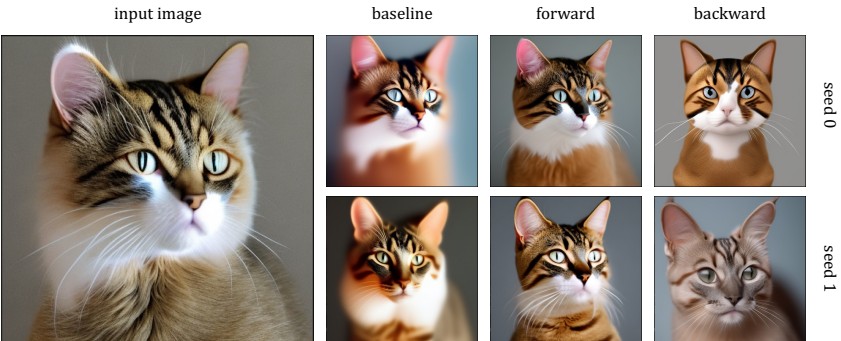

Figure 7: **Comparison of different simulated annealing variations.** The baseline finds noises that follow the pose but suffer from blurriness, while the backward fails to capture the content of the image well. We found that the forward loss works better across various examples.

Since ZCA whitening is the whitening operator that maximizes the cross-covariance between the input image and the target noise distribution, this observation empirically supports the observation that diffusion models learn to map noise to images while preserving as much as possible the correlation between them (see Figure 2).

## 5  NOISE OPTIMIZATION THROUGH SIMULATED ANNEALING

We uncovered the existence of a strong correlation between noise samples and the images they generate in diffusion models. Moreover, previous experiments demonstrated that an image $\mathbf{z}_0$ is related to the corresponding solution of Equation (6) through a linear transform that is close to ZCA whitening when $t \simeq T$. Properly blending the "noise" $\boldsymbol{\epsilon}^*(\mathbf{z}_0, t)$ and the $\mathbf{z}_0$ image and passing it through the diffusion model recovers the solution of Equation (6). This implies that the estimated final image from step $t$ matches exactly the input image in one step starting at $t$, i.e. $\mathbf{z}_{0|t} = \mathbf{z}_0$, and a single diffusion step is enough to generate the final image.

Because whitening is invertible, the reverse mapping is also well-defined: a one-step inference starting from a random Gaussian noise sample effectively maps to the final estimated image, which is a *de*-whitened version of the noise itself. This is an insightful trait, as it means we can obtain an approximate estimated final image $\mathbf{z}_{0|T}$ by simply de-whitening the input noise with a matrix multiplication, *without using any diffusion model*.

We leverage this discovery to create *model-agnostic correlated noises*. Given an input image $\mathbf{x}$, we propose to find a noise $\mathbf{z}$ that correlates with it by minimizing a loss subject to the noise being in-distribution, i.e.

$$\mathbf{z} = \arg\min_{\mathbf{z}} \mathcal{L}(\mathbf{z}, \mathbf{x}) \qquad \text{s.t.} \quad \mathbf{z} \sim \mathcal{N}(0, \mathbf{I}) \tag{10}$$

Equation (10) can be solved by simply searching for a noise that is correlated with an image. In this way, we can compute noises that generate very similar images to the target image across various models.

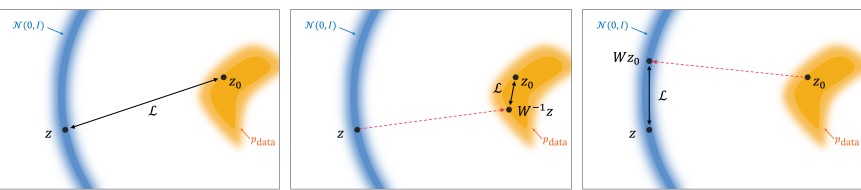

Figure 8: **Noise search with Simulated Annealing.** We consider three implementations of simulated annealing: baseline (a), forward (b), and backward (c).

---

**Algorithm 1:** Simulated Annealing for Correlated Noise Search

---

**Input:** Target image $\mathbf{x}$, Number of iterations $N$, Initial temperature $T_{\text{init}}$, Final temperature $T_{\text{final}}$, Step size parameter $\alpha$
**Output:** Best noise $\mathbf{z}_{\text{best}}$
**Initialize** $\mathbf{z}_{\text{cur}} \leftarrow \mathbf{z}_{\text{init}}$, $L_{\text{cur}} \leftarrow \mathcal{L}_{\text{fwd}}(\mathbf{z}_{\text{cur}}, \mathbf{x})$
$\mathbf{z}_{\text{best}} \leftarrow \mathbf{z}_{\text{cur}}$, $L_{\text{best}} \leftarrow L_{\text{cur}}$
**for** $k \leftarrow 1$ **to** $N$ **do**
    $T_k \leftarrow T_{\text{init}} + (T_{\text{final}} - T_{\text{init}}) \times \frac{k}{N}$
    Sample random noise $\boldsymbol{\epsilon} \sim \mathcal{N}(0, 1)$;
    $\mathbf{z}_{\text{new}} \leftarrow \sin\left(\frac{\pi}{2}\alpha\right)\mathbf{z}_{\text{cur}} + \cos\left(\frac{\pi}{2}\alpha\right)\boldsymbol{\epsilon}$;
    $L_{\text{new}} \leftarrow \mathcal{L}_{\text{fwd}}(\mathbf{z}_{\text{new}}, \mathbf{x})$
    **if** $L_{new} < L_{cur}$ **then**
        $\mathbf{z}_{\text{cur}} \leftarrow \mathbf{z}_{\text{new}}$, $L_{\text{cur}} \leftarrow L_{\text{new}}$;
    **else**
        **if** $rnd() < \exp\left(-\frac{L_{new} - L_{cur}}{T_k}\right)$ **then**
            $\mathbf{z}_{\text{cur}} \leftarrow \mathbf{z}_{\text{new}}$, $L_{\text{cur}} \leftarrow L_{\text{new}}$;
        **end**
    **end**
    **if** $L_{new} < L_{best}$ **then**
        $\mathbf{z}_{\text{best}} \leftarrow \mathbf{z}_{\text{new}}$, $L_{\text{best}} \leftarrow L_{\text{new}}$;
    **end**
**end**
**Return** $\mathbf{z}_{\text{best}}$

---

**Simulated Annealing** is a gradient-free optimization method (Kirkpatrick et al., 1983) that samples a new candidate solution at each iteration, ensuring that the optimized variable stays in distribution while simultaneously matching a given energy target. New candidates for next iterations are generally computed within a certain range of the current iteration. In our application, we propose using spherical interpolation to sample a neighboring noise as

$$\mathbf{z}_{k+1} = \sin\left(\frac{\pi}{2}\alpha\right)\mathbf{z}_k + \cos\left(\frac{\pi}{2}\alpha\right)\boldsymbol{\epsilon} \tag{11}$$

where $\boldsymbol{\epsilon} \sim \mathcal{N}(0, \boldsymbol{I})$ is a randomly sampled Gaussian white noise image, and $\alpha \in [0, 1]$. Spherical interpolation ensures the resulting noise remains Gaussian of variance 1. Different variations of the $L_2$ loss are considered in Equation (10) and demonstrated visually in Figure 8. These are:

- **Baseline**: we compare directly the noise with the target image: $\mathcal{L}_{\text{base}}(\mathbf{z}, \mathbf{x}) = \|\mathbf{z} - \mathbf{x}\|^2$
- **Forward**: we approximate the first step of a diffusion model with a ZCA de-whitening operation and compare with the image: $\mathcal{L}_{\text{fwd}}(\mathbf{z}, \mathbf{x}) = \|\mathbf{W}_{\text{ZCA}}^{-1}\mathbf{z} - \mathbf{x}\|^2$. This requires performing a matrix multiplication at every step.
- **Backward**: we reproject the target image with the whitening matrix and compare to the noise: $\mathcal{L}_{\text{bwd}}(\mathbf{z}, \mathbf{x}) = \|\mathbf{z} - \mathbf{W}_{\text{ZCA}}\mathbf{x}\|^2$. In this case a single multiplication is required at the start with the target image.

Figure 7 shows images generated by different instances of the $L_2$ loss. The baseline loss $\mathcal{L}_{\text{base}}$ is able to follow the content of the original image but produces blurry results, while the backward $\mathcal{L}_{\text{bwd}}$ loss fails to faithfully reproduce an image with similar content as the original. The forward loss $\mathcal{L}_{\text{fwd}}$ is the one that produces best results, and its the one that we employed to generate other examples in this paper. The algorithm is described in Algorithm 1.

## 6 RESULTS AND APPLICATIONS

**Image variation generation.** Given an input image and a prompt that roughly describes it, our goal is to find initial Gaussian noises that could generate variations of the image when denoised by a diffusion model through DDIM sampling. Figure 1 (top left) defines the initial image to be matched.

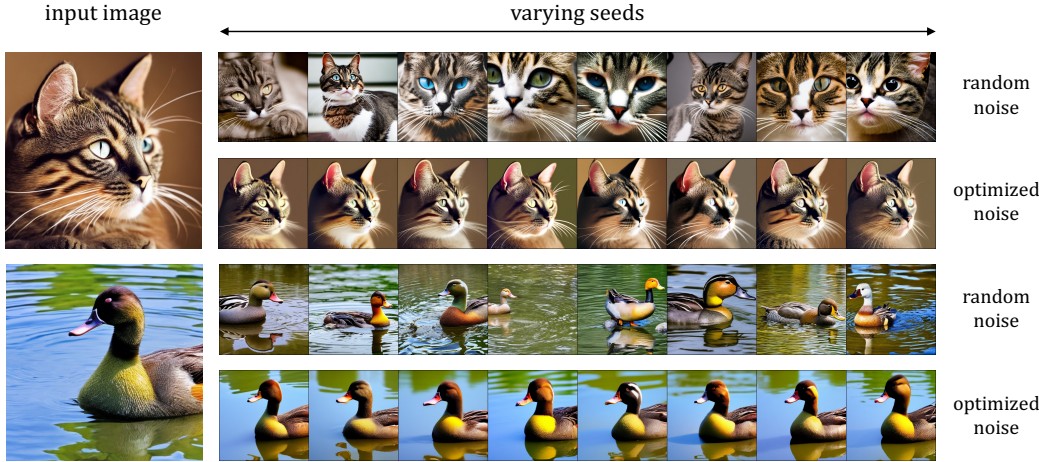

input image                    varying seeds

Figure 9: **Model-free image variation generation.** Starting from random noises that generate diverse outputs, our noise sampling based on simulated annealing converges to a set of noises which generate variations of the target image.

We employ simulated annealing to optimize Equation (10), generating a batch of noises without any diffusion model involved (top row). We then show that the same noise is able to generate similar content even when employing different models such as Stable Diffusion 2.1, Stable Diffusion 1.5 and SD-Turbo. Figure 9 and different results included in the Appendix demonstrate the generalization capacity of our approach.

**Prompt-based image editing.** We consider the task of editing an image by partially reverting it to noise and denoising it with a different prompt. SDEdit (Meng et al., 2022) adds random noise to the original image before denoising it. As more noise is added, less structures from the original image are preserved during editing. On the other hand, DDIM inversion can be efficiently used for editing. Similarly, one can partially invert the DDIM sampling process up to an intermediate step, and denoise with a new prompt. In this case, the result is deterministic. By employing our noise sampler to obtain noise that is correlated with our image, we can improve the structure preservation in SDEdit, allowing us to go to higher noise levels while simultaneously better preserving the content of the image. In Figure 10, we compare SDEdit and DDIM inversion on an example task that replaces a cat with a rabbit.

# 7 RELATED WORK

**Whitening transforms in machine learning.** Whitening transforms can be employed to normalize and decorrelate input data, and served as inspiration for batch normalization (Ioffe & Szegedy, 2015). ZCA whitening can be particularly interesting when applied to activation functions, which can improve generalization, speed-up training and approximate better features Luo (2017); Huang et al. (2018). However, when indiscriminately applied to data without normalized gradient descent methods (Grosse & Martens, 2016), data whitening can hinder generalization (Wadia et al.; Ahmad, 2024). These works demonstrate that whitening can provide effective insights into learning dynamics, though its application in diffusion models remains underexplored.

**Noise analysis in diffusion models.** Understanding how images progressively deteriorate with noise and how that affects diffusion models is paramount to improve accuracy and speed-up training (Lee et al., 2022; Chen et al., 2024; Huang et al., 2024), facilitate temporal coherency (Ge et al., 2023; Chang et al., 2023), and elucidate the analysis of noise parameters (Deasy et al., 2022; Jolicoeur-Martineau et al., 2023). Different works (Xu et al., 2024; Mao et al., 2024) have studied the effect of random seeds in generative models, underscoring the sensitivity of generative processes to noise configurations. Moreover, taking into consideration noise correlations with itself (Ge et al., 2023; Chang et al., 2023) and with samples on the data distribution (Mao et al., 2024; Xu et al., 2024) can be useful to improve the training and evaluation of diffusion models. Lastly, alternative corruption schemes (Bansal et al.; Daras et al., 2022; Bansal et al., 2023; Rissanen et al., 2023; Hoogeboom &

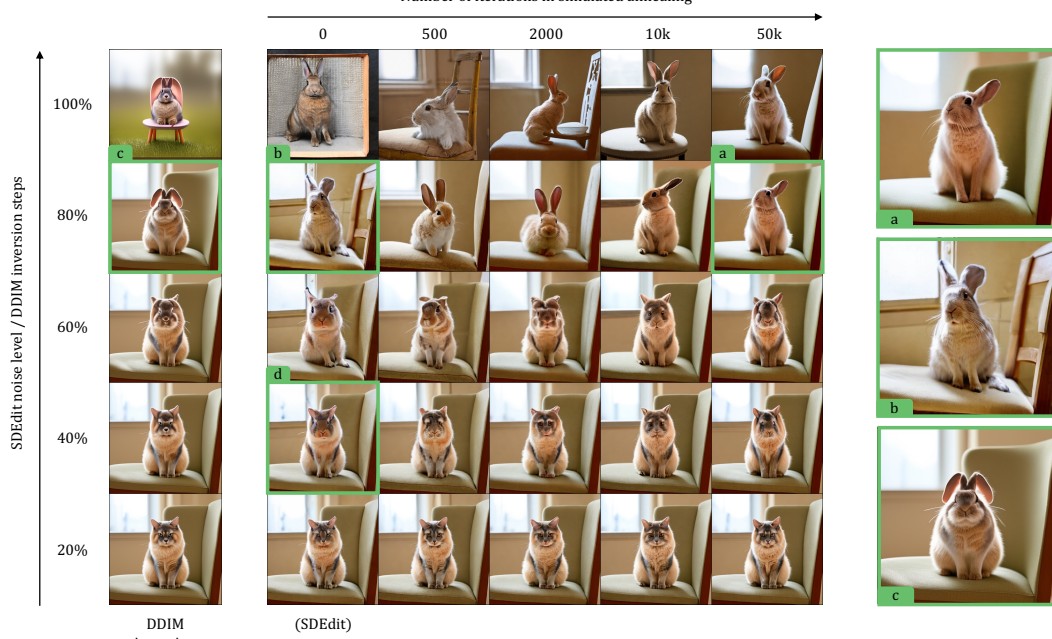

Figure 10: **Improving SDEdit with our noise.** By adding a correlated noise found with our algorithm when doing SDEdit, we are able to preserve the overall structure of the image better while allowing realistic changes (a), while traditional SDEdit loses many of the original structure at high noise level (b). At low noise level, the content is too strong and forces the rabbit to fit the shape of the cat (d). A similar observation can be made for DDIM inversion (c).

Salimans, 2024) demonstrate that besides noise, other operations that can progressively deteriorate images can be also used in diffusion training.

**Inverse problems and noise optimization.** Optimizing the generation process for image-space objectives is a recent trend in diffusion literature. Inverse problems can be solved by guiding the diffusion process through a function that compares the image-space objective with the predicted Tweedie for a given timestep (Kadkhodaie & Simoncelli, 2021; Lugmayr et al., 2022; Yu et al., 2023; He et al., 2023; Song et al., 2023; Chung et al., 2024a;b). A more accurate approach compares the result of the diffusion model through a fixed computational graph that is fully differentiated through noise optimization (Samuel et al., 2023; Wallace et al., 2023; Pan et al., 2023; Hong et al., 2023; Ben-Hamu et al., 2024; Eyring et al., 2024; Samuel et al., 2024). While this approach can produce more accurate results, its also computationally more expensive, since it requires the computation of the gradients of the objective function relative to the noise. For a survey on inverse problems with diffusion models, please refer to Daras et al. (2024).

## 8 CONCLUSIONS

In this work, we focused on the correlation between noise and corresponding images in diffusion models. To this extend, we proposed to study the optimization problem of finding a "noise" that would minimize the fixed-time score matching objective for a given target image. This gave us new insights on the distribution mapping learned by the diffusion model. Importantly, we empirically demonstrated that diffusion models learn to perform ZCA de-whitening in the setting of single-step inference from noise. Leveraging this interesting discovery, we proposed to replace a diffusion evaluation by a simple matrix multiplication and use this to find model-agnostic noises, which can generate image variations and help image editing across diverse models.

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

# A APPENDIX

## A.1 ADDITIONAL RESULTS

We provide additional examples showing the effect of using the sampled noise from our method. In Figure 11 we recover different sampled noises using our approach based on the input image and generate images for different prompts, showcasing its potential use for editing. In Figures 12 and 13 we combine our method with SDEdit. All the results are generated using Stable Diffusion 2.1.

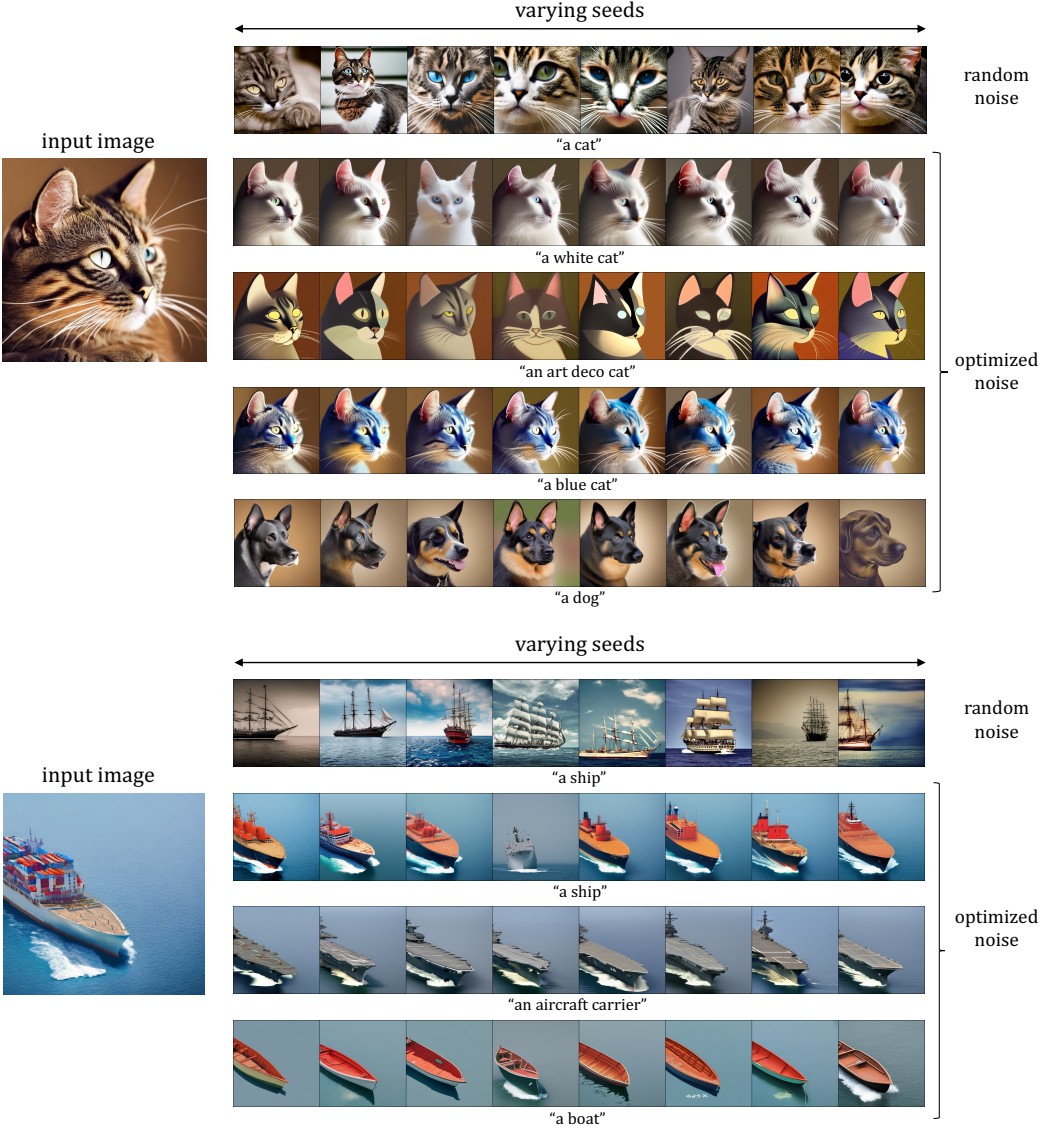

Figure 11: **Model-free image variation generation.** Our method searches for noises that are correlated with the target image *without using any diffusion models*. This allows us to generate image variations that preserve the original structure by using different prompts.

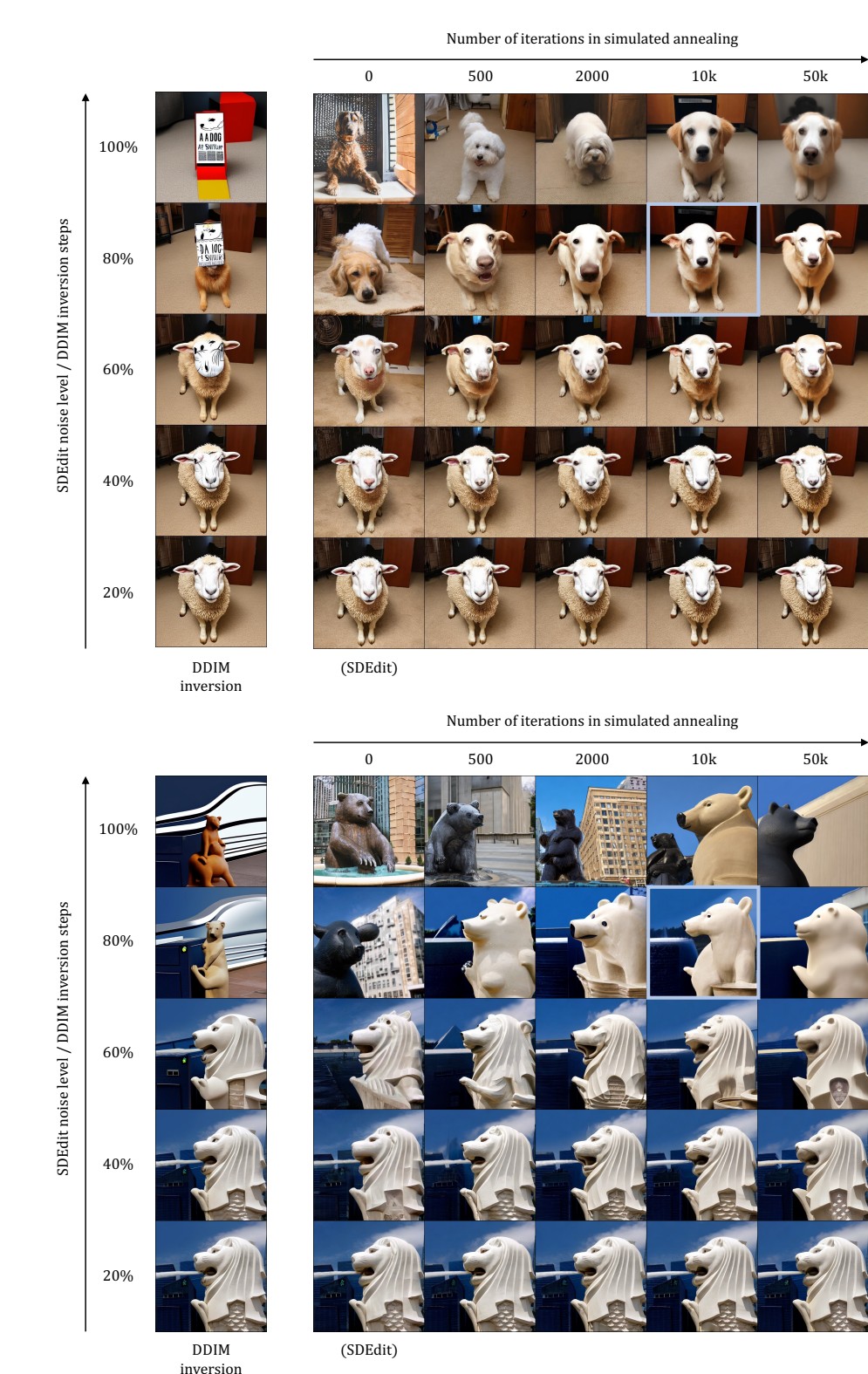

Figure 12: **Improving SDEdit with our noise.** Our sampled noise can help to preserve the structure of the original image, even in situations where DDIM inversion can fail.

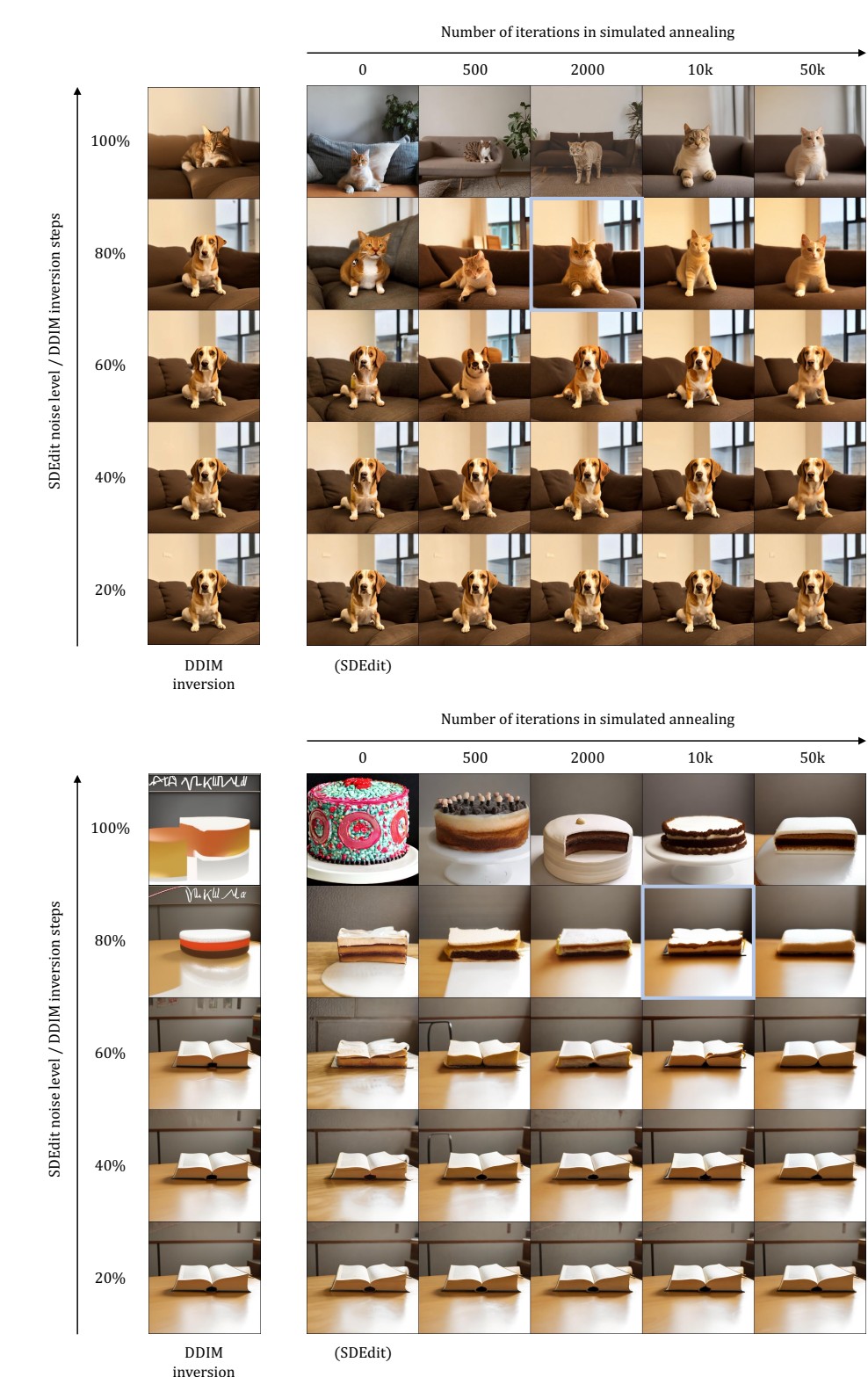

Figure 13: **Improving SDEdit with our noise.** Our sampled noise can help to preserve the structure of the original image, even in situations where DDIM inversion can fail.

