# OpenReview forum: "First-Step Inference in Diffusion Models Learns Image De-whitening"
_ICLR.cc/2025/Conference — ICLR 2025 Conference Withdrawn Submission_

### Official Review · Reviewer_1CLo · 2024-10-21

**Soundness:** 2
**Presentation:** 3
**Contribution:** 2
**Rating:** 3
**Confidence:** 5

**Summary:**

This paper explores the correlation between input noise and generated images in diffusion models, aiming to reveal how diffusion models maintain the relationship between noise and images during the denoising process. Specifically, the study proposes an approximation of a single-step mapping through fixed-point inference (first-step inference) and finds that this mapping closely aligns with the ZCA de-whitening transform. The experimental results demonstrate that the single-step inference achieved through noise optimization closely aligns with the ZCA de-whitening transform. The effectiveness of this linear mapping was validated on the ImageNet dataset, showing that the optimized noise can generate consistent image variations across different models. Additionally, the optimized noise improved structural preservation in image editing tasks, maintaining the overall content of the image even at high noise levels, outperforming traditional methods such as SDEdit.

**Strengths:**

- It was discovered that the initial denoising operation of diffusion models can be approximated by the ZCA de-whitening transform, revealing the global structure that associates noise with images in the model.
- Noise optimization was achieved using the simulated annealing algorithm, enabling the ability to generate similar images across multiple models.
- The optimized noise was shown to improve the performance of image editing methods such as SDEdit, better preserving image structure at high noise levels.
- The optimized noise can be applied across different diffusion models, enhancing the generalizability of the approach.
- This method outperforms traditional approaches in preserving image structure at high noise levels, increasing the flexibility of image editing.

**Weaknesses:**

- Using simulated annealing for noise optimization requires multiple iterations, affecting efficiency.
- The effectiveness of ZCA de-whitening depends on the data distribution, which may limit the model's performance on unseen datasets.
- The approach is based on observed assumptions without providing a rigorous analysis.

**Questions:**

1.  About Hypothesis 1:
- Equation (8) is the most critical finding (or core contribution) of this paper. Is this discovery purely based on observation? What is the motivation?
- Since it only holds approximately when $ T $ is large, it seems that there exists an upper bound for the gap that is independent of $ \epsilon_{\theta} $?
- Why was $ t = 0.98T $ chosen for the experiment? Could $ t = T $ be useful instead?
- For the different models $ \epsilon_{\theta_1}, \epsilon_{\theta_2}, \dots, \epsilon_{\theta_n}$, if the assumption holds, the optimal solution should be $ \epsilon^*_{\theta_1} \approx \epsilon^*_{\theta_2} \approx \dots \approx \epsilon^*_{\theta_n} $ for the same $z_0$, which seems counterintuitive. This suggests that different models yield the same solution for the same $ x_t $ regarding Equation (6), even though $ \epsilon_{\theta_1}(x_t, t), \epsilon_{\theta_2}(x_t, t), \dots, \epsilon_{\theta_n}(x_t, t) $ are expected to differ.

2. SDEdit performs best at 40%-60% timesteps, which seems to contradict the hypothesis in the paper that $ t $ needs to be very large. Does this pose a conflict?

3. For inversion-based methods, this paper significantly improves upon the original DDIM inversion method. However, some existing approaches [1,2,3] have already improved the inversion reconstruction loss, achieving more precise consistency. What are the advantages of this method compared to those? However, it seems that this method cannot achieve complete reconstruction, although it can maintain consistency within a certain range. Therefore, I would like to know how effective this method is for image editing in complex scenarios, such as those with rich backgrounds or multiple objects—specifically, whether it can maintain background consistency.

4. Could you compare the results of directly adding random noise to $z_0$ to obtain $z_t $, then denoising back to $z_0$? Perhaps randomly adding noise and then denoising might also achieve good results, as $z_t$ would still contain information from $z_0$ in this case.



### Reference:

[1] Cho H, Lee J, Kim S B, et al. Noise map guidance: Inversion with spatial context for real image editing. ICLR 2024.

[2] Xu S, Huang Y, Pan J, et al. Inversion-free image editing with natural language. CVPR 2024.

[3] Ju X, Zeng A, Bian Y, et al. Pnp inversion: Boosting diffusion-based editing with 3 lines of code. ICLR 2024.

---

### Official Review · Reviewer_DkqQ · 2024-10-27

**Soundness:** 2
**Presentation:** 2
**Contribution:** 1
**Rating:** 3
**Confidence:** 4

**Summary:**

This paper makes two main contributions:

1. The first step of the sampling process of diffusion models (or single-step approximation of the full sampling trajectory) can be modeled using image de-whitening techniques, particularly ZCA.
2. Through fixed-point iteration, it is possible to find noise corresponding to an image, which shows the ability to generate similar images given different diffusion models, and improvements in image editing methods, specifically SDEdit.

**Strengths:**

- The qualitative results applied to SDEdit are somewhat promising.

**Weaknesses:**

**1. Replication of Previous Work**

The first downside of this paper is that its contributions, especially those validated through experiments, have already been claimed in existing literatures. The paper experimentally demonstrates that via fixed-point iteration, we can identify noise corresponding to an image, which can be used to (1) generate similar images across different models and (2) assist with image editing using SDEdit.

However, regarding point (1), there are already results showing that if the noise is the same, similar images are generated even with different models.
- *The Emergence of Reproducibility and Generalizability in Diffusion Models* ([ICML24](https://arxiv.org/abs/2310.05264))

Moreover, research has already proposed finding noise through fixed-point iteration and using it for editing in various ways. In particular, this approach has also been applied to image editing. Besides the papers I listed, I recall other examples using fixed-point techniques.
- *On Exact Inversion of DPM-Solvers* ([CVPR24](https://arxiv.org/abs/2311.18387))
- *ReNoise: Real Image Inversion Through Iterative Noising* ([ECCV24](https://arxiv.org/abs/2403.14602))
- *Lightning-Fast Image Inversion and Editing for Text-to-Image Diffusion Models* ([Arxiv23](https://arxiv.org/abs/2312.12540))

Additionally, the lack of any quantitative metrics for the experimental results is also an issue.

**2. Overclaim**

The second issue with this paper is overclaiming their argument with insufficient experimental results, especially when they claim that ZCA de-whitening and the first step of diffusion models are similar. The key to verifying this claim lies in choosing a de-whitening method that resembles the diffusion model. However, in my opinion, the notion that ZCA is the most similar among de-whitening methods is quite different from the claim that the first step of the diffusion model can be understood as ZCA de-whitening. For example, we already understand the latent code of diffusion model as an optimal transport solution [1]. Why do you think the framework of ZCA de-whitening gives us better understanding of the diffusion models? Can you validate that ZCA de-whitening is better theory to understand the diffusion models?

[1] : Understanding DDPM Latent Codes Through Optimal Transport ([ICLR23](https://openreview.net/forum?id=6PIrhAx1j4i))

If the theoretical contribution were significant, the paper could still be evaluated positively even if the empirical contribution is small (Weakness #1), but this does not seem to be the case here.

**Questions:**

- Are there advantages to using simulated annealing (SA) over gradient-based (GD) optimization? I want to know the qualitative difference between using SA and GD.

---

### Official Review · Reviewer_UNrE · 2024-10-31

**Soundness:** 3
**Presentation:** 2
**Contribution:** 2
**Rating:** 5
**Confidence:** 4

**Summary:**

This paper explores the intricate relationship between input noise and generated images. Specifically, it finds that the initial denoising step performed by the network can be approximated as image de-whitening (ZCA). Consequently, the paper proposes a model-agnostic method for sampling correlated noises. Finally, it discusses two applications of this phenomenon.

**Strengths:**

1. This phenomenon (the initial inference closely resembles ZCA) is interesting.
2. This phenomenon is observed in many diffusion models
3. The authors conduct multiple experiments to investigate this phenomenon.
4. This paper demonstrates that the first-step inference approximates a linear transformation and does not depend on the model. Consequently, it proposes a model-agnostic method.
5. The paper identifies two applications for this finding, where the prompt-based image editing is useful.

**Weaknesses:**

1. While this phenomenon is interesting, its potential applications may be quite limited, as it only holds true for the first step. Although you have identified two applications, one of them—image variation generation—is not widely discussed.
2. Although this phenomenon is interesting, it may not be particularly amazing, as any non-linear function can be approximated by a linear function within a small interval.
3. Focusing solely on linear operations related to whitening is too narrow in scope. Although you provide a motivation in Figure 4 indicating that the results of Equation 6 bear a striking resemblance to the effects of ZCA whitening, this does not imply that only whitening should be considered. I believe there are many other linear transformations worth discussing. For instance, the identity transformation may also yield good performance, as suggested by the experiments in Section 4.

**Questions:**

1. Can you provide additional applications for your discovery?
2. Can you offer stronger evidence to demonstrate that ZCA is the best approximation? Perhaps comparing it with more commonly used linear transformations would be better.

---

### Official Review · Reviewer_iSYq · 2024-11-03

**Soundness:** 3
**Presentation:** 3
**Contribution:** 2
**Rating:** 6
**Confidence:** 4

**Summary:**

The paper suggests that the first step of a diffusion model is similar the dewightning using ZCA. It shows that it is much more correlated with ZCA than other dewightning approaches like PCA and others. Then it search for the best noise to use to generate similar images to a given one, i.e., try to perform noise inversion, by simply correlating the noise after dewithning with the target image. They show it can be useful to perform editing on one example.

**Strengths:**

The correlation to ZCA shown is done while comparing to other alternatives.
The editing experiment is nice
The different demonstration shown throughout the paper are quite nice
The paper is interesting.

**Weaknesses:**

The first part of the work is nice and the experiments done are quite rigorous showing why ZCA and not other options. Yet, the second part of the editing and simulated annealing is quite trivial and not really convincing. Basically checking each time what happens after one step of denoising and if it is similar to the original image is expected to lead to the results shown. Moreover, the fact that the results are demonstrated on few images only is very limited. It feels like strong cherry picking. Also, there are many other inversion methods. In addition, one may apply the same correlation with just simple denoising.

**Questions:**

1. Not sure I understand Figure 1. I don't see any correlation between the different rows. In SD1.5 all the cats look to the right. In SD Turbo you see head rotation, which is not present in the previous two rows.
2. You write "We hypothetize that the gap between the fitted one (Diff) and (ZCA) might be partly due to the fact that the ZCA
whitening matrix was only estimated on a subset of ImageNet, while the fitted one would reflect the entire training distribution of the diffusion model". You can easily check this hypothesis by simply both increasing and decreasing the size of the data used to calculate ZCA and see if it increases and reduces  the gap, respectively.
3. Why you show the inversion experiment just on few images? Feels like strong cherry picking
4. The fact that the learning of the noise is correlated is not surprising. There are many works that show that the diffusion process learn different level of details throughout the diffusion and that it is not just learning Gaussian noise. The claim in the paper that we would expect learning Gaussian noise in the optimization in (6) is not well justified
5. Any real theory for why ZCA?

---

### Official Review · Reviewer_LHDb · 2024-11-04

**Soundness:** 3
**Presentation:** 3
**Contribution:** 2
**Rating:** 5
**Confidence:** 3

**Summary:**

The paper analyzes the correlation between noise and the images generated through DDIM sampling, showing that the one-step approximation of the DDIM inversion noise for any given image closely relates to the Zero-phase Component Analysis (ZCA) inverse whitening transform applied to that image. Based on this observation, the paper proposes a simple yet effective simulated annealing method to identify correlated noises, demonstrating its utility in tasks such as image variation generation and editing.

**Strengths:**

1. The paper is well-written, clear, and easy to follow. The proposed idea is well-motivated, simple, and effective. It begins by introducing the observed phenomenon that noise and images generated by DDIM are correlated, followed by a well-supported hypothesis, demonstrated through detailed analysis.
2. The simulated annealing algorithm for correlated noise proves useful for image variation generation and editing tasks, yielding decent generation quality.

**Weaknesses:**

The main weakness of the paper lies in the lack of quantitative comparisons and discussions regarding existing baseline methods, making it challenging to objectively assess the performance advantages of the proposed approach. Specifically:

1. There is no performance and efficiency comparison between the proposed model-agnostic method and other commonly used DDIM inversion techniques, leaving a gap in understanding the practical advantages in real-world applications.

2. While SDEdit with correlated noise visually preserves more structural similarity compared to random noise, the paper only provides qualitative results. Although the method appears effective, the absence of comprehensive quantitative comparisons hinders a full evaluation of its performance.

**Questions:**

Please see weaknesses part.

---

### Official Review · Reviewer_tfxS · 2024-11-04

**Soundness:** 4
**Presentation:** 3
**Contribution:** 2
**Rating:** 5
**Confidence:** 4

**Summary:**

The paper explores how the initial noise-to-image mapping of diffusion models, particularly with deterministic DDIM sampling and ZCA image whitening.
Through optimizing the noise with a fixed-point iteration and simulated annealing approach, the method preserves the structure of the original image at noise levels. The author further apply the proposed method to improving image editing.

**Strengths:**

- The paper is well-written, and easy to follow.
- The authors extend their analysis to real-world applications, such as image editing, and shows promising results.
- The analyzation of noise and image with image whitening operator is quite novel.

**Weaknesses:**

- Authors show the correlation of image and noise, however it is not quite novel. Since DDIM is deterministic, same noise initialization to any score based generative model with same training objective will yields same image. (For instance, fig1 with DDIM inversion will yield similar results.)
- With respect to the analyzation, the author empirically found that image whitening operation to noise space. It would be better if there was a more mathematically proven explanation, since the main concern of the paper is related to the analysis of the strictly mathematical model. For example, what is the mathematical reason why hypothesis 1 in the diffusion model actually holds? This should be thoroughly explained in section 3 or in the appendix.
- With respect to the application, where is the quantitive results? I understand that it is not easy to quantitively evaluate in the image editing, however author can evaluate quantitively through experiments in SDEdit.

In summary, the analysis by image whitening is novel, but the paper contains only empirical motivation and quantitative results. It would be a better paper if the above weakness were addressed.

1. Su, Xuan, et al. "Dual Diffusion Implicit Bridges for Image-to-Image Translation." The Eleventh International Conference on Learning Representations.
2. Hur, Jiwan, et al. "Expanding Expressiveness of Diffusion Models with Limited Data via Self-Distillation based Fine-Tuning." Proceedings of the IEEE/CVF Winter Conference on Applications of Computer Vision. 2024.

**Questions:**

See the weakness.

In addition, author insists "efficient optimization algorithm". What is the actual computation cost, or searching time compared to the other baseline?

---

### Author Response · Authors · 2024-12-03

We sincerely thank the reviewers for their valuable comments and constructive suggestions. After carefully considering the feedback, we have identified two key areas for improvement: (1) the need for a stronger theoretical foundation to support our claim connecting diffusion models with ZCA, and (2) the necessity of a more rigorous quantitative evaluation of the editing application presented in the second part of the paper. In light of these observations, we have decided to withdraw our submission and will incorporate these insights to refine and strengthen our work for future iterations.

---

### Note · Authors · 2024-12-03

I have read and agree with the venue's withdrawal policy on behalf of myself and my co-authors.